# Metallization of 3D-Printed PET and PETG Samples with Different Filling Densities of the Inner Layers

**DOI:** 10.3390/ma18143401

**Published:** 2025-07-20

**Authors:** Sonya Petrova, Diana Lazarova, Mihaela Georgieva, Maria Petrova, Dimiter Dobrev, Dimitre Ditchev

**Affiliations:** 1Department of Chemistry, Technical University of Sofia, 1000 Sofia, Bulgaria; mggeorgieva@tu-sofia.bg; 2Department of Electrochemistry and Corrosion, Institute of Physical Chemistry, Bulgarian Academy of Science, “Academic Georgi Bonchev” Street, Blok 11, 1113 Sofia, Bulgaria; 3Institute of Mathematics and Informatics, Bulgarian Academy of Sciences, “Academic Georgi Bonchev” Street, Blok 8, 1113 Sofia, Bulgaria; 4Department of Railway Engineering, Technical University of Sofia, 1000 Sofia, Bulgaria

**Keywords:** 3D printing, FDM technology, polymer samples, PET, PETG, electroless deposition, copper layers, nickel layers, conductive layers

## Abstract

The aim of the study was to develop a suitable pre-treatment (and more specifically, the etching operation) of 3D-printed PET and PETG samples with different filling densities of the inner layers for subsequent electroless metallization. The influence of temperature, etching time, and sodium hydroxide concentration in the etching solution on the deposition rate, adhesion, and composition of Ni-P coatings was determined. The studies show that a high temperature and concentration of the etching solution do not improve the properties of the coating. The etching not only plays an important role in improving adhesion but also affects the composition and thickness of the nickel layer. It was also established how the degree of filling densities of the inner layers affects the uniformity, penetration depth, and thickness of electrolessly deposited Cu and Ni-P coatings on 3D PETG samples.

## 1. Introduction

In recent years, the three-dimensional (3D) printing of samples from different types of materials has quickly become a regular operation in a number of sectors, e.g., the military and aviation industry, industrial design, automotive, electronics, and medicine [1].

The most common technologies used for 3D printing are multi-material jetting photopolymer 3D printing (PolyJet), stereolithography (SLA), direct metal laser sintering (DMLS), fused deposition modeling (FDM), selective laser sintering (SLS) and aerosol jet printing (AJP) [2,3,4,5,6,7,8].

FDM technology is becoming increasingly important in the field of additive manufacturing. The technique was originally developed for rapid prototyping, allowing for the testing and evaluation of products during the design phase [9]. It is considered an easy and environmentally friendly method, and its greatest achievement is the ability to make objects with complex geometry with multiple cavities and holes. In this technology, the modeling process is carried out by overlay, extruding heated thermoplastic material layer by layer through a precision nozzle. FDM technology is the only additive technology by which thermoplastic products with excellent mechanical, thermal, and chemical characteristics can be made [3,10,11].

Important for the applicability of samples obtained by 3D technology is the possibility of metallization. With the advent of rapid prototyping, metallization on polymers is attracting attention in a variety of technological applications, from electronic devices and automobiles to prosthetics. Green technologies exist that are able to metallize selected areas of 3D-printed samples by introducing PdCl_2_ as a catalytic precursor into the polymer filament for surface activation and subsequent metallization. The methods were also developed for electroless metal deposition on polymer with Cu particles embedded in it [12,13,14,15]. A process for the electroless metallization of plastics was developed back in the 1960s. By this method, a thin metallic layer (most often copper or nickel) is deposited, which can then be thickened by the electrodeposition of a coating of the same or another metal. In the 1970s, this process became common practice in the automotive industry in order to reduce the weight of parts, and in the 1980s, it quickly spread to electronics and other industries [16,17,18].

Three-dimensionally printed products are subjected to metallization, most often those made from acrylonitrile butadiene styrene (ABS) [1,19], polylactic acid (PLA) [10,20], polyethylene terephthalate (PET) [21], glycol-modified polyethylene terephthalate (PETG) [1,22,23], etc.

One of the most affordable thermoplastics is ABS. This material is lightweight, flexible, and durable, and can be easily extruded. A disadvantage of using it as an incandescent 3D printer is that it produces concentrated fumes during printing, which can be harmful to people with respiratory conditions.

PET has been used for decades in the production of food bottles and containers as a substitute for PVC, in the production of synthetic fibers for clothing, and, in recent years, as a 3D printing material. It is a thermoplastic amorphous polymer with good mechanical properties, excellent impact resistance, and moderate resistance in a highly alkaline environment at room temperature, but it degrades at high temperatures. A solution of 96% sulfuric acid causes the destruction of the polymer. The metallization of conventional flexible PET (polyester textile) samples has been investigated in detail [16,24,25,26,27,28]. It has been found that staining PET samples with a potassium or sodium base leads to the formation of caverns on the polymer surface, which leads to an increase in the thickness of the deposited metal coating [25,27].

PETG is an amorphous plastic that combines the properties of PET and glycol. The addition of glycol reduces the overheating effect of PET and reduces brittleness. On the other hand, this material combines both the strength of ABS and the ease of PLA 3D printing. This makes it an impressive printing filament and one of the best materials for 3D printing, characterized as it is by excellent hardness, flexibility, strength, resistance to chemicals and to impact, transparency, elasticity, excellent thermal stability, and easy extrusion. It is biocompatible and is used in the manufacture of food packaging, ensuring their safety [22,23]. PETG is an ideal material for the 3D printing of objects that can be subjected to sudden loads, e.g., mechanical components, parts of 3D printers, and other security features. Some of the advantages of using PETG filament in FDM technology are as follows:(1)There is excellent adhesion between layers;(2)PETG fingerprints do not warp or shrink easily;(3)PETG can be recycled along with its footprints.

The combination of electroless metallization and the rapid prototyping of polymeric materials by 3D printing is extremely important in the development of modern technology, as it produces “composite materials” of the dielectric/metallic (conductive) surface type, which are characterized by improved appearance, functionality, and durability, and would find applications in a variety of fields: in cosmetic packaging and blister packaging for photosensitive drugs in medicine, to give a shiny, mirror, or chrome effect that makes the product more attractive; in electronics, to protect against electromagnetic interference in, e.g., casings for electronic components or displays; in the automotive industry, to imitate metal while being lightweight, impact-resistant and cheaper; and in medicine, to give additional sterility and resistance to UV/acids. Moreover, PETG is biocompatible, and metallization makes it even more resistant to aggressive environments; it can also be used for decorative and protective elements in the interior (panels, moldings, etc.).

The aim of this study was to develop an appropriate pre-treatment (and, in particular, the etching operation) of 3D-printed PET and PETG samples with different filling densities of the inner layers for subsequent electroless metallization. It was also important to establish how the filling rate affects the uniformity, depth of penetration, and thickness of the electrolessly deposited Cu and Ni-P coatings.

## 2. Experimental Part

Samples of 3D PET and 3D PETG with dimensions of 10 mm × 10 mm × 2 mm; filling densities of the inner layers of 20%, 50%, and 100%; and a resolution of 0.8 mm were obtained by 3D printing (SKYMAKER A2, Sky-Tech Taiwan Electronics Co., Ltd., Jhonghe, Taiwan) using FDM technology (Figure 1a–c).

The 3D patterns were obtained by transforming solid thermoplastic filaments (PET or PETG) into a molten state and sequentially adding layer by layer of molten material using a computer-controlled extrusion head until the desired structure was formed upon solidification. The first, “bottom” polymer layer is applied to a preheated base and has a smoother pattern than the last “top” layer. The different pattern of the two sides of the samples determines the different properties of these surfaces.

The 3D-printed samples were electrolessly metallized according to the scheme presented in Figure 2.

Pre-treatment includes the following operations (Figure 2a):(1)Simultaneous degreasing and etching in an alkaline medium. The influence of working conditions was investigated within the following ranges: NaOH 100 ÷ 400 g L^−1^, temperature 40 °C ÷ 70 °C, and treatment duration 3 ÷ 30 min.(2)Pre-activation in a solution of 3 M HCl, T = 25 ± 2 °C, and a time of 3 min.(3)Activation in a colloidal Pd/Sn activating solution: activator-A-75-12 (TU-Sofia), T = 25 ± 2 °C, and a time of 5 min.(4)Acceleration in an alkaline medium: T = 25 ± 2 °C and a time of 5 min.

The processed samples were metallized in solutions for the electroless deposition of Ni-P or Cu coatings (Figure 2b), the compositions and operating conditions of which are given in Table 1.

The number of experiments we have performed is between 5 and 10. The results obtained are very close and within the measurement error.

The morphology of the samples was observed/examined by scanning electron microscopy (SEM) (JSM 6390, JEOL, Tokyo, Japan), and the elemental composition was determined by Energy Dispersive Spectroscopy (EDS) (Oxford Instruments INCA x-sight, Aztec Software, Oxford, UK). The phase composition of the coatings was characterized by powder X-ray diffraction (XRD), with a vertical diffractometer Philips PW 1050 and a secondary monochromator, working with CuKα radiation (Philips Sourced, Almelo, The Netherlands). The thickness of the chemical coatings was measured by the gravimetric method, as well as by X-ray fluorescence analysis (XRFA) (Fischerscope X-RAY XDAL, HELMUT FISCHER GMBH, Institut fur Elektronik und Messtechnik, Sindelfingen, Germany).

The adhesion of the coatings was determined by two methods—with *a standard test with adhesive tape* [29] and *a quantitative direct pull-off test* [30]. The latter uses brass checkers glued with acrylic glue to the metal coating. The applied force for the separation of the metal layer from the sample is calculated in kg ms^−2^ [20].

## 3. Results and Discussion

### 3.1. Effect of the Etching Operation from Pre-Treatment of 3D Printed PET and PETG Samples

One of the main preliminary operations in the electroless plating of dielectric materials is etching. Solutions containing sulfuric and chromic acid are effective for this purpose in many cases. However, using such a solution for 3D-printed samples, uneven Ni-P coatings with poor adhesion were obtained. Taking into account the harmfulness of chromium sulfuric acid solution to the environment, subsequent studies were focused on the possibility of replacing it with one containing NaOH. Bernasconi et al. found that chemical modification occurs simultaneously with the alkaline etching of the polymer surface, where new hydroxyl and carboxyl functional groups (–OH and –COOH) are formed. This leads to an increase in robbing, creating micro- and nanostructures on the surface and thus improving the mechanical anchoring of the metallic coating and the adhesion of the deposited metallic coating, and hence ensuring better electroless metallization [29,31,32].

The subject of this study was 3D-printed samples of PET and PETG with a filling density of the inner layers of 100%. The influence of the composition and working conditions of the etching solutions on the thickness, elemental composition, and adhesion of the electrolessly deposited Ni-P coatings was investigated. The influence of the concentration of NaOH in the etching solution on the adhesion of the Ni-P coating and its thickness is shown in Figure 3.

When etching the 3D PET samples, it was observed that with an increase in the concentration of NaOH to 200 g L^−1^, there was almost no change in the thickness and content of the included phosphorus in the coating (Ni 87.61 wt.%; P 7.98 wt.%). At twice the concentration of the NaOH (400 g L^−1^), the thickness of the coating, as well as the phosphorus content (Ni 82.03 wt.%; P 3.69 wt.%), decreased. This may be due to the fact that 3D PET is resistant in highly alkaline environments, resulting in the insufficient micro-roughness of the surface of the specimens. Through the direct draw test, it was found that the best adhesion was in cases of staining in 100 g L^−1^ NaOH.

For the 3D PETG samples, the inverse relationship was observed compared to that of the 3D PET samples. When using an etching solution containing 100 g L^−1^ NaOH, the content of included phosphorus in the coating was P 6.96 wt.% (Ni 75.28 wt.%), the adhesion was *Х* = 2.49 kg ms^−2^, and the thickness of the Ni-P coating was 6.95 μm. Increasing the concentration of NaOH to 400 g L^−1^ led to the increased adhesion of the Ni-P coating (*Х* = 2.82 kg ms^−2^) due to better robbing and better catalysis of the polymer surface, which increased the chemical reaction rate; the thickness of the Ni-P coating (8.05 μm) and the amount of included phosphorus (Ni 89.49 wt.%; P 9.94 wt.%) also increased (Figure 3).

From the obtained experimental data on the influence of the concentration of NaOH in the etching solution, it was found that when metallizing 3D PET samples, it is recommended to work with 100 g L^−1^ NaOH, whereas for 3D PETG samples, a more concentrated solution containing 400 g L^−1^ NaOH is needed.

The concentration of NaOH in the staining solution also influences the final morphology of the deposited Ni-P coatings, which can be seen in Figure 4. At the lower concentration of NaOH (100 g L^−1^), where the robbing is less, the coating has a more grainy and uniform structure. This can be seen in Figure 4a. The luminous dots could be individual crystallites that grow more intensely in the direction perpendicular to the surface and are recorded as such, as they are the most prominent. When the NaOH concentration was increased to 400 g L^−1^ (Figure 4b), the surface was more deeply pitted, leading to better surface catalysis and resulting in higher chemical reaction rates and a more intense deposition of electroless Ni-P and hydrogen. The porous structure indicates that hydrogen was released locally, in preferential zones (such as larger Pd grains), by forming pores.

The influence of operating conditions in the etching operation—treatment time (Figure 5) and temperature (Figure 6)—on the adhesion and thickness of the Ni-P coating was also investigated.

The obtained data on the etching time show that in both types of 3D-printed samples, increasing the treatment time to 15 min leads to an increase in the thickness of the deposited coatings, the content of the deposited P (from Ni 86.95 wt.% and P 6.98 wt.% (at τ = 3 min) to Ni 87.61 wt.% and P 7.98 wt.% (τ = 15 min)), and the adhesion of the coatings. With increasing the etching time to 30 min, no significant change in thickness is observed, and the adhesion of the deposited Ni-P coating deteriorates.

From Figure 6, it can be seen that the greatest thickness and adhesion of the coating on 3D samples is obtained at T = 60 °C. An increase in the temperature above T = 60 °C leads to a decrease in the thickness of the coating and a deterioration in adhesion, which is due to a slight “over-etching” effect of the polymer surface.

Based on the data obtained for the thickness, elemental composition, and adhesion of the Ni-P coatings, the optimal etching time (15 min) and the optimum temperature (60 °C) for both materials were determined.

### 3.2. Electroless Deposition of Cu and Ni-P Coatings on 3D PETG Samples with Different Filling Densities of the Inner Layers (20%, 50% and 100%)

These studies were conducted with the 3D PETG samples due to their greater strength, durability, impact resistance, and ability to operate at higher temperatures compared to PET.

With the established operating conditions of the etching solution (400 g L^−1^ NaOH, T = 60 °C, and τ = 15 min), studies were carried out to establish the influence of the degree of filling density of the inner layers on the thickness, elemental composition and adhesion of Cu and Ni-P coatings (Table 2).

The data in this table shows that with an increase in the structural zone (20% → 100%), the deposition rate and the coatings’ thickness increase, which is also confirmed by the higher content of Cu, Ni, and P, respectively, according to the EDS analysis. The low coating thickness values at the lower filling density of the inner layers (20%), on the other hand, are due to the looser structure, which is due to the “over-etching” effect and leads to a decrease in the deposition rate.

From the phosphorus content, it can be concluded that Ni-P coatings have a mixed to amorphous structure [17]. The globular structure of the surface of the coatings is visible from the SEM images (Figure 7).

The adhesion of the obtained samples was determined by two methods—a standard test with adhesive tape [Type-Test-Method ASTM D 3359-83] and a quantitative direct pull-off test [direct pull-off test (DPO) according to the ASTM D 4541-02 Standard] (Table 2). The results of both methods confirm the good adhesion of the Ni-P coatings.

Studies were carried out to establish the phase composition of the electroless Ni-P and Cu coatings on the 3D PETG samples with different filling densities of the inner layers. The diffraction patterns of the deposited Cu and Ni-P coatings are presented in Figure 8.

In both cases, there is a clearly pronounced signal from the coatings. For the copper layers (Figure 8a), four copper peaks, namely, (111), (200), (220), and (311), are observed, and it should be noted that in all three samples, there is a plane orientation with Miller indices (111). The larger the filling of the 3D structure of PETG, the less pronounced the orientation of the deposited coating. The peak positions best match the reference card PDF code 00-004-0836 from the ICDD database (The International Centre for Diffraction Data), equivalent to the ICSD (Inorganic Crystal Structure Database) collection code 43493.

In the Ni-P coatings, a very wide single diffraction line appears, which is located approximately around the position of the Ni diffraction line with Miller indices (111). As the P content increases, the peak at 44.80 *2θ* remains significantly widened. Similar results for coatings containing up to 10 wt.% P were obtained by Z. Guo et al. [33]. The type of nickel coatings strongly depends on their phosphorus content. According to Rolf Weil and Konrad Parker [34], as well as other researchers, microcrystalline coatings are obtained at phosphorus contents below 7%. When higher amounts of phosphorus are included, amorphous coatings are formed. In the samples we obtained, the amount of phosphorus depends on the electrolyte used, the deposition conditions, and the preliminary preparation. The least amount of phosphorus is included when the substrate is etched in high concentrations of NaOH (3.6 wt.%). At the deposition conditions we selected as optimal, the amount of phosphorus ranged from 8.87 wt.% to 9.94 wt.%. At these values, the deposited coatings are generally amorphous. Interestingly, in our samples, besides an amorphous halo, small peaks of nickel are also visible. The presence of nickel can be explained by the fact that the shape of the substrates is complex, and, in some places, the access of phosphorus has been limited, as a consequence of which we have founde little nickel.

The SEM images in Figure 9 and Figure 10 show the penetration of the Cu and Ni-P coatings in a diagonal section of the 3D PETG specimens with varying degrees of filling density of the inner layers. With an increase in the filling rate of the inner layers (20 → 50%), the compaction of the internal cavities and an increase in the thickness of the coatings are observed.

## 4. Conclusions

Studies have been carried out on the electroless metallization of the 3D-printed samples of PET and PETG with different filling densities of the inner layers (20%, 50%, and 100%). The samples were processed according to the following scheme: degreasing and etching, activation in a colloidal palladium–tin solution, acceleration, and subsequent electroless metallization.

From the studies carried out for the etching operation, it was found that an environmentally harmful etching solution containing Cr^6+^ can be successfully replaced with one containing an optimal concentration of NaOH of 100 g L^−1^ (for 3D PET) and 400 g L^−1^ (for 3D PETG) at a temperature of 60 °C and an etching time of 15 min.

The results obtained for the electrolessly deposited Ni-P and Cu coatings on the 3D-printed samples from PETG show that with an increase in the structural zone from 20% to 100% of the filling density of the inner layers, there is an increase in the deposition rate and the thickness of the electrolessly deposited layers, which is a consequence of the compaction of the internal cavities.

Uniform Ni-P and Cu coatings on 3D PETG samples, with the greatest thickness, good adhesion, and the highest percentage of inclusion of Ni (89.49 wt.%) and Cu (84.39 wt.%), are obtained at 100% filling density of the inner layers. The phosphorus content of the Ni-P coatings determines their X-ray amorphous character.

## Figures and Tables

**Figure 1 materials-18-03401-f001:**
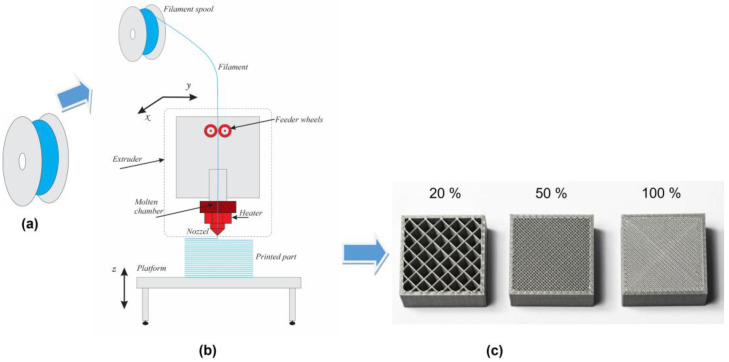
Scheme of the 3D-printing method using FDM technology: (**a**) a roll with a thermoplastic polymer filament, (**b**) an FDM printing platform, and (**c**) 3D-printed samples with varying degrees of filling density of the inner layers.

**Figure 2 materials-18-03401-f002:**
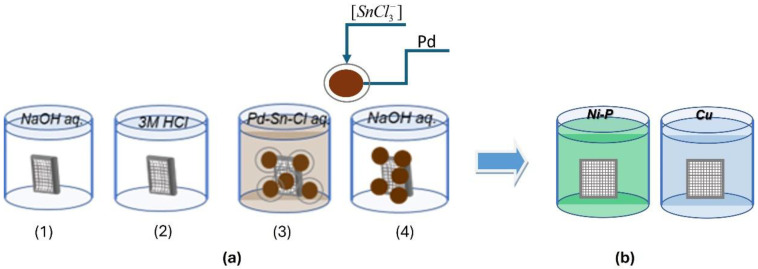
Scheme of operations for pre-treatment (**a**) and electroless metallization (**b**).

**Figure 3 materials-18-03401-f003:**
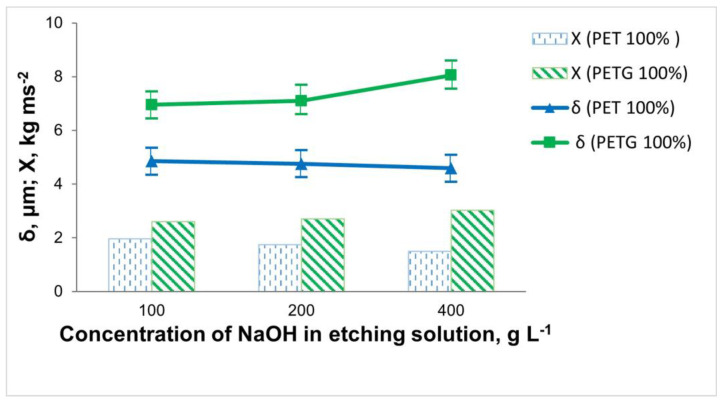
Dependence of the thickness and adhesion of the Ni-P coatings on the 3D samples of PET and PETG on the concentration of NaOH in the etching solution (at T = 60 °C and τ = 15 min).

**Figure 4 materials-18-03401-f004:**
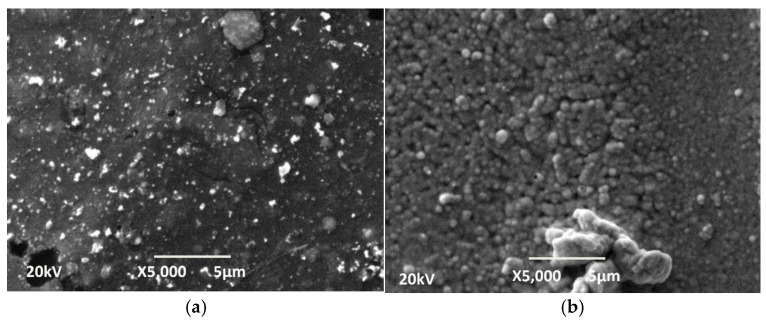
SEM photographs of the Ni-P coatings on samples of 3D PETG etching for 15 min in (**a**) 100 g L^−1^ NaOH and (**b**) 400 g L^−1^ NaOH.

**Figure 5 materials-18-03401-f005:**
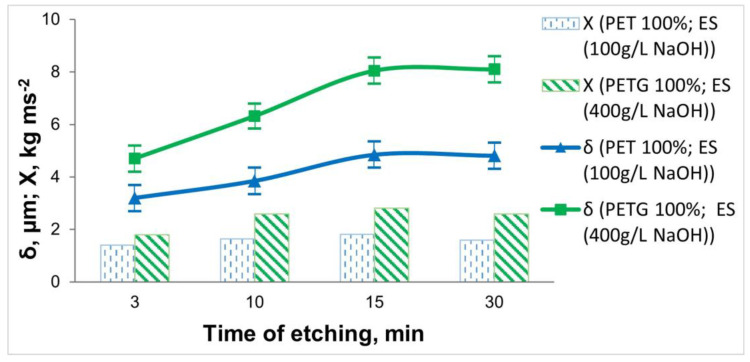
Dependence of the thickness and adhesion of the Ni-P coatings on the 3D PET and PETG samples on the etching time (at T = 60 °C).

**Figure 6 materials-18-03401-f006:**
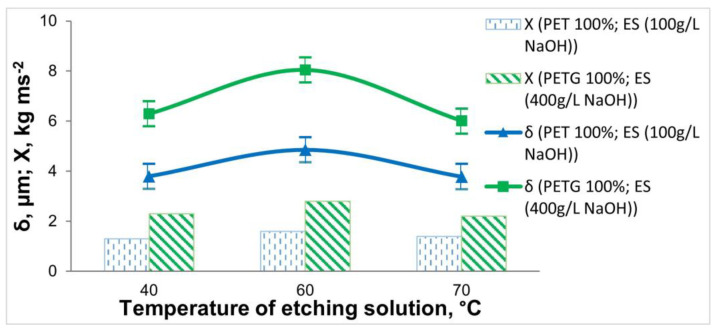
Dependence of the thickness and adhesion of the Ni-P coatings on the 3D samples of PET and PETG on the temperature of the etching solution (at τ = 15 min).

**Figure 7 materials-18-03401-f007:**
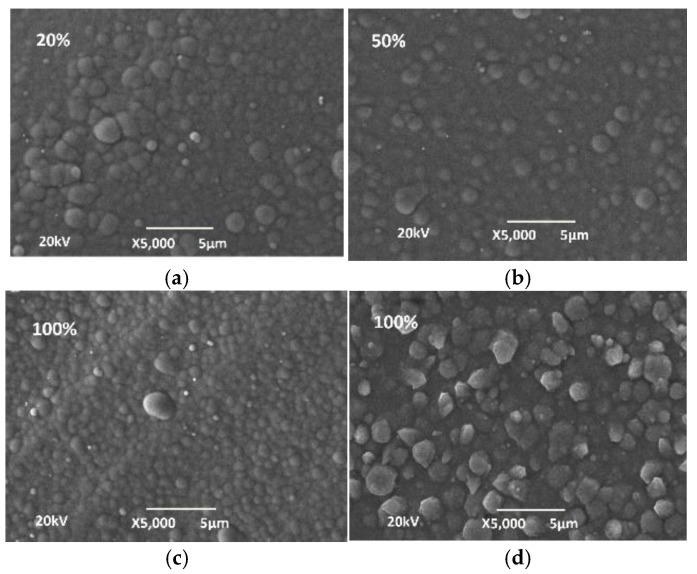
SEM images of 3D PETG with varying degrees of filling density of the inner layers and electrolessly deposited (**a**,**c**) Ni-P coatings and (**d**) Cu coatings: (**а**) with a filling density of 20%; (**b**) a filling density of 50%; and (**c**,**d**) a filling density of 100%.

**Figure 8 materials-18-03401-f008:**
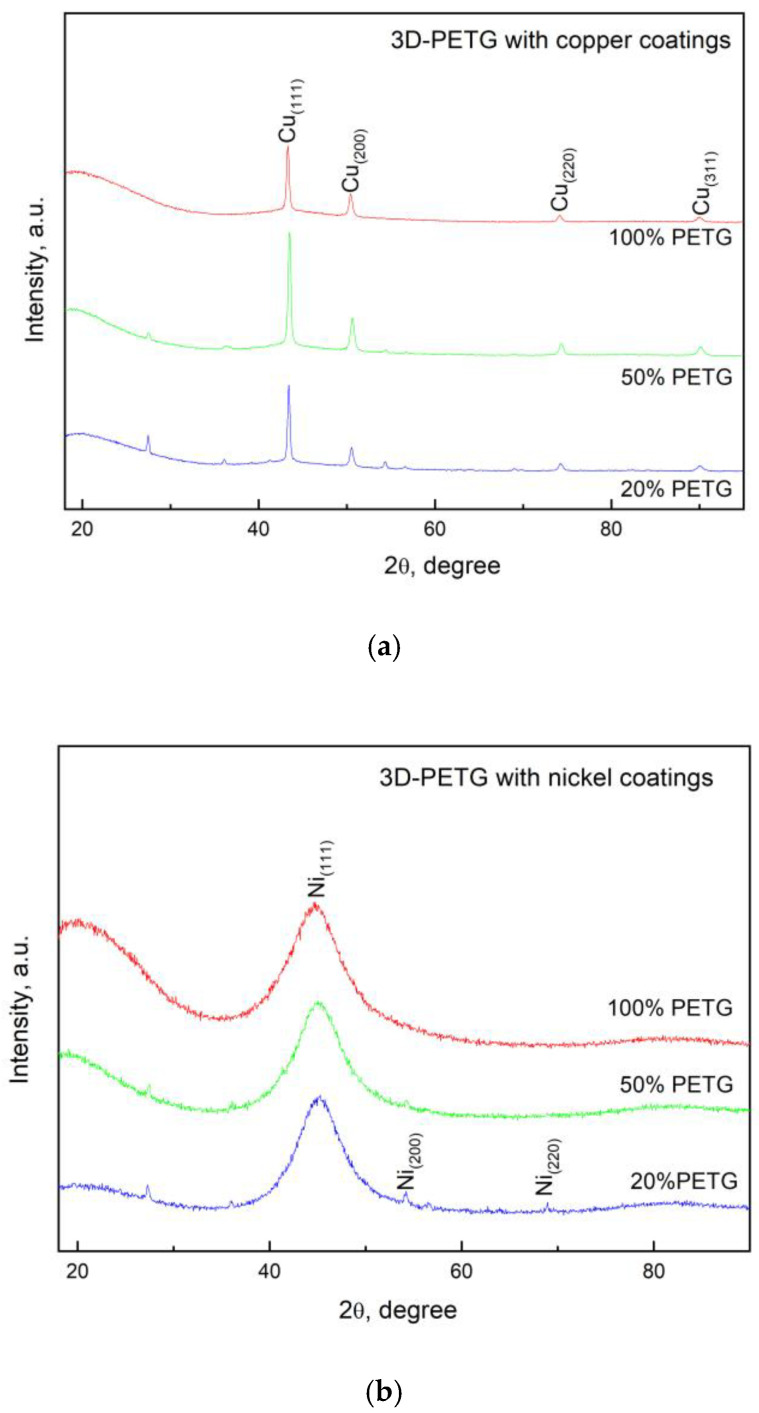
XRD diffractograms of the 3D PETG samples with varying filling densities of the inner layers and electrolessly deposited (**a**) Cu coatings and (**b**) Ni-P coatings.

**Figure 9 materials-18-03401-f009:**
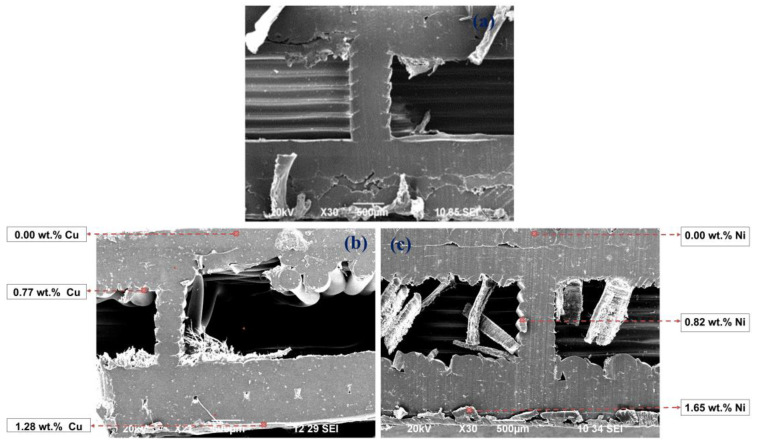
SEM images of the diagonal section of 3D PETG samples with 20% filling density of the inner layers: untreated (**a**) and electrolessly metallized with a Cu coating (**b**) and a Ni-P coating (**c**).

**Figure 10 materials-18-03401-f010:**
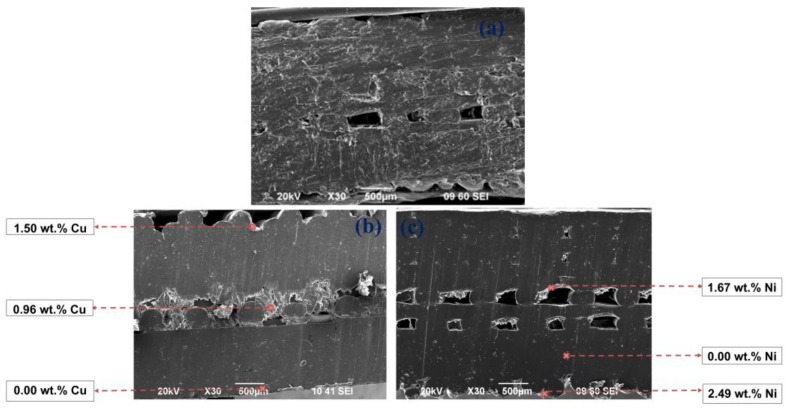
SEM images of the diagonal section of the 3D PETG samples with 50% filling density of the inner layers: untreated (**a**) and electrolessly metallized with a Cu coating (**b**) and a Ni-P coating (**c**).

**Table 1 materials-18-03401-t001:** Composition and working conditions of the solutions for electroless plating.

Composition and Working Conditions	Ni-P Bath	Cu Bath
CuSO_4_.5H_2_O, g L^−1^		10
NiSO_4_.7H_2_O, g L^−1^	25	
Na_2_-EDTA, g L^−1^		40
NaH_2_PO_2_.H_2_O, g L^−1^	22	
CH_2_O, mL L^−1^		10
C_5_H_6_O_3_, mL L^−1^	17.4	
NaOH, g L^−1^		10
CH_3_COONa.3H_2_O, g L^−1^	20	
T, °C	82 ± 2	45 ± 2
Ph	4.6 ÷ 4.8	12.8 ÷ 13.0
Time, min	30	

**Table 2 materials-18-03401-t002:** Influence of the filling density of the inner layers of 3D-PETG samples on the thickness (***δ***), adhesion (***X***), and elemental composition of the deposited Cu and Ni-P coatings.

Electroless Coatings		Cu			Ni-P		
Filling Density		20%	50%	100%	20%	50%	100%
***δ*, µm**	**Gravimetrical**	1.29	1.57	2.46	5.82	7.58	8.05
	**XRF**	1.91	1.92	1.98	4.93	5.99	6.20
***X*, kg ms^−2^**		0.43	0.96	1.05	1.64	2.74	2.82
**EDS, wt.%**	**Cu**	81.98	83.21	84.39			
	**Ni**				83.86	89.01	89.49
	**P**				8.87	9.04	9.94

## Data Availability

The original contributions presented in this study are included in the article. Further inquiries can be directed to the corresponding authors.

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
