# Peer review of "Metallization of 3D-Printed PET and PETG Samples with Different Filling Densities of the Inner Layers"

_materials, 2025, doi:10.3390/ma18143401_

Round 1

Reviewer 1 Report

Comments and Suggestions for Authors

The author reported a systematic study on electrodeless metallization of FDM printed PET and PETG products with different internal filling densities. They displayed the optimization of etching during the surface modification. Different parameters were optimized to get the best results including the concentration of NaOH, etching time and temperature. This study is well-aligned with the functionalization of FDM printed products. It provided an effective way to achieve uniform metal coating on the printed plastic parts with good bonding. This paper is recommended for publication after solving the following questions.

  1. The author reported different trends about the coating thickness and adhesion between PET and PETG products in response to the concentration change of NaOH. More detailed explanations should be provided to explain these differences instead of just briefly mentioning the possible reason-surface roughness. The author should provide more details/evidence about the roughness change of the PET and PETG.
  2. Figure 4 shows an obvious morphology change in the Ni coating at different NaOH concentrations. SEM images show one sample with continuous film and another with porosity structure. The Author should explain the reason of such an obvious morphology change. Are these relating to etching rates, nucleation density or change of surface energy or something else.
  3. The practical implications of this work should be discussed. And why the part needed to be surface conductive. What are the potential applications, especially combining the fast-prototyping capabilities of the 3D printing.
  4. XRD of the Ni coating shows the diffraction peak of Ni (111) plan and some other weak peak. But the author mentioned that “The phosphorus content of Ni-P coatings determines their X-ray amorphous character” in the conclusion, It seem this is contradict with the result from the XRD. Please comment on this. The author also needs to clarify the observed broadening peak of the Ni. Is this because of the nanoscale crystallinities, mixed-phase structures or the true amorphous characteristics.
  5. In the introduction part about the 3D printing technology. Aerosol jet printing is another commonly used method. Aerosol jet printing is design and famous for the printing of electronics and sensors. The author should also mention these in the introduction. For reference, some of the papers about Aerosol jet printing can be read and cited. Mechanism and application: Nature, 617, 292–298 (2023). Advanced Materials, 2025, 37, 2414203. The printing of electronics and sensors: ACS Applied Nano Materials, 7 (8), 9453–9459;  Small, 20 (5), 2304966.
  6. The manuscript would benefit a lot from a thorough English language and format checking for clarity and fluency. There are numerous grammar problems. For example, Electrodeless plating instead of electrodeless plaiting. The unit of temperature should be °C instead of 0 The unit of XRD is not correct. And the SEM images are missing scale bar.

Author Response

Comments 1: The author reported different trends about the coating thickness and adhesion between PET and PETG products in response to the concentration change of NaOH. More detailed explanations should be provided to explain these differences instead of just briefly mentioning the possible reason-surface roughness. The author should provide more details/evidence about the roughness change of the PET and PETG.

Response 1: Thank you for pointing this out. We agree with this comment. [Bernasconi et al have found that chemical modification occurs simultaneously with alkaline etching of the polymer surface, where new hydroxyl and carboxyl functional groups (-OH and -COOH) are formed. This leads to an increase in robbing, creating micro- and nanostructures on the surface and thus improving the mechanical anchoring of the metallic coating and better adhesion of the deposited metallic coating, and hence better electroless metallization [29, 31-32]. – Results and discussion, page 5, paragraph 1]

Comments 2: Figure 4 shows an obvious morphology change in the Ni coating at different NaOH concentrations. SEM images show one sample with continuous film and another with porosity structure. The Author should explain the reason of such an obvious morphology change. Are these relating to etching rates, nucleation density or change of surface energy or something else.

Response 2: [For 3D PETG samples, the inverse relationship was observed to that of 3D PET samples. When using an etching solution containing 100 g L-1 NaOH, the content of included phosphorus in the coating is P 6.96 wt%, respectively Ni 75.28 wt%, the adhesion is Ð¥=2.49 kg ms-2 and the thickness of Ni-P coatings is 6.95 μm. Increasing the concentration of NaOH to 400 g L-1, leads to increased adhesion of Ni-P coating (Ð¥=2.82 kg ms -2), due to better robbing and better catalysis of the polymer surface, which increases the chemical reaction rate, respectively the thickness of Ni-P coatings (8.05 μm) and the amount of including phosphorus (Ni 89.49 wt%; P 9.94 wt%) increases (Fig. 3).  The concentration of NaOH in the staining solution also influences the final mor-phology of the deposited Ni-P coatings, which can be seen in Fig. 4. At the lower concentration of NaOH (100 g L-1) where the robbing is less, the coating has a more grainy and uniform structure. This can be seen in Fig. 4a. The luminous dots could be individual crystallites that grow more intensely in the direction perpendicular to the surface and are recorded as such as they are the most prominent. When the NaOH concentration was increased to 400 g L-1 (Fig. 4b), the surface was more deeply pitted, leading to better surface catalysis and resulting in higher chemical reaction rates, more intense deposition of electroless Ni-P and of hydrogen. The porous structure indicates that hydrogen was released locally, on preferential zones (such may be larger Pd grains) by forming pores. - Results and discussion, page 6, paragraphs  2 and 4.]

Comments 3: The practical implications of this work should be discussed. And why the part needed to be surface conductive. What are the potential applications, especially combining the fast-prototyping capabilities of the 3D printing.

Response 3: Thank you for pointing this out. We agree with this comment. [The combination of electroless metallization and rapid prototyping of polymeric materials by 3D printing is extremely important in the development of modern technology, as it produces “composite materials” of the dielectric/metallic (conductive) surface type, which are characterized by improved appearance, functionality and durability, and would find applications in a variety of applications: to give a shiny, mirror or chrome effect that makes the product more attractive - cosmetic packaging, blister packaging for photosensitive drugs in medicine; in electronics it is used to protect against electromagnetic interference - casings for electronic components or displays; in the automotive industry - imitates metal, but it is lightweight, impact-resistant and cheaper; in medicine - gives additional sterility and resistance to UV/acids; PETG is biocompatible, and metallization makes it even more resistant to aggressive environments; used for decorative and protective elements in the interior (panels, moldings, etc.). – Introduction, page 3, paragraph 2]

Comments 4: XRD of the Ni coating shows the diffraction peak of Ni (111) plan and some other weak peak. But the author mentioned that “The phosphorus content of Ni-P coatings determines their X-ray amorphous character” in the conclusion, It seem this is contradict with the result from the XRD. Please comment on this. The author also needs to clarify the observed broadening peak of the Ni. Is this because of the nanoscale crystallinities, mixed-phase structures or the true amorphous characteristics.

Response 4: Thank you for pointing this out. We agree with this comment. [The type of nickel coatings strongly depends on their phosphorus content. According to Rolf Weil and Konrad Parker [34] and other researchers have shown that at phosphorus contents below 7% microcrystalline coatings are obtained. When higher amounts of phosphorus are included, amorphous coatings are formed. In the samples we obtained, the amount of phosphorus depends on the electrolyte used, the deposition conditions and the preliminary preparation. The least amount of phosphorus is included when the substrate is etched in high concentrations of NaOH (3.6 wt.%). At the deposition conditions we selected as optimal, the amount of phosphorus ranged from 8.87 wt.% to 9.94 wt.%. At these values the deposited coatings are generally amorphous. Interestingly in our samples, besides amorphous halo, small peaks of nickel are also visible. The presence of nickel can be explained by the fact that the shape of the substrates is complex and in some places the access of phosphorus has been limited as a consequence of which we have correlated little nickel. – Results and discussion, page 10, paragraph 2]

Comments 5: In the introduction part about the 3D printing technology. Aerosol jet printing is another commonly used method. Aerosol jet printing is design and famous for the printing of electronics and sensors. The author should also mention these in the introduction. For reference, some of the papers about Aerosol jet printing can be read and cited. Mechanism and application: Nature, 617, 292–298 (2023). Advanced Materials, 2025, 37, 2414203. The printing of electronics and sensors: ACS Applied Nano Materials, 7 (8), 9453–9459; Small, 20 (5), 2304966.

Response 5: Thank you for pointing this out. We agree with this comment and the references cited in the text on Introduction, page 2, paragraph 2. [5. Liu, B.; Liu, S.; Devaraj, V.; Yin, Y.; Zhang, Y.; Ai, J.; Feng, Y.H.J. Metal 3D nanoprinting with coupled fields, Nature Communications, 2023, 14, 4920.; 6. Song, K.; Zhou, J.; Wei, C.; Ponnuchamy, A.; Bappy, M.O.; Liao, Y.; Jiang, Q.; Du, Y.; Evans, C.J.; Wyatt, B.C.; O’Sullivan, T.; Roeder, R.K.; Anasori, B.; Hoffman, A.J.; Jin, L.; Duan, X.; Zhang, Y. A Printed Microscopic Universal Gradient Interface for Super Stretchable Strain-Insensitive Bioelectronics, Advanced Materials, 2025, 37 (11), 2414203.; 7. Bappy, M.O.; Jiang, Q.; Atampugre, S.; Zhang, Y. Aerosol Jet Printing of High-Temperature Bimodal Sensors for Simultaneous Strain and Temperature Sensing Using Gold and Indium Tin Oxide Nanoparticle Inks, ACS Applied Nano Materials, 2024, 7 (8), pp. 9453-9459.; 8. Du, Y.; Reitemeier, J.; Jiang, Q.; Bappy, M.O.; Bohn, P.W.; Zhang, Y. Hybrid Printing of Fully Integrated Microfluidic Devices for Biosensing, Small, 2023, 20 (5), 2304966]

Comments 6: The manuscript would benefit a lot from a thorough English language and format checking for clarity and fluency. There are numerous grammar problems. For example, Electrodeless plating instead of electrodeless plaiting. The unit of temperature should be °C instead of 0 The unit of XRD is not correct. And the SEM images are missing scale bar.

Response 6: Thank you for pointing this out.

We agree with this comment and the remark “°C” has been corrected in the text.

Regarding the XRD unit of the presented diffraction patterns, we only compare them with each other by offsetting them by intensity for this purpose. For this reason, it would be unfair to use counts or counts per second and we use the dimensionless arbitrary unit (a.u.).

The SEM images in Figures 9 and 10 have been replaced with scale bar images - page 11.

Reviewer 2 Report

Comments and Suggestions for Authors

The manuscript is devoted to the development of a suitable pretreatment (more precisely, etching) of 3D-printed PET and PETG samples with different internal layer filling densities for subsequent electroless metallization. In my opinion, the manuscript addresses a relevant issue and contains interesting results. The manuscript provides a detailed, step-by-step description of the experiments and their discussion. The conclusions are supported by the results obtained.

I have the following comments on the manuscript:

  1. Experimental part. “…elemental composition was determined by En-ergy Dispersive Spectroscopy (EDS).” No detector name is available.
  2. Figure 3. The graph does not contain confidence intervals. It is important to understand the difference in values.
  3. Figure 5-6. Confidence intervals are also not shown on the graphs.
  4. Table 2. Here, authors must also provide confidence intervals for measured values, including for EDS analysis.
  5. Figure 7. The captions for the figures are misaligned.
  6. Figure 7. SEM images of 3D PETG with varying degrees of filling density of the inner layers and electroless deposed: (a-c) Ni-P coatings; (d) Cu coating.” Here, authors must also identify captions (a)-(c) for fill densities of 20%, 50%, and 100%.
  7. Figures 9, 10. There are no scale marks on SEM images of microstructures.

Author Response

Comments 1: Experimental part. “…elemental composition was determined by En-ergy Dispersive Spectroscopy (EDS).” No detector name is available.

Response 1: The remark has been noted and the EDS type (Oxford Instruments INCA x-sight) has been added on page 5.

Comments 2: Figure 3. The graph does not contain confidence intervals. It is important to understand the difference in values.

Response 2: We agree with this comment. We have changed the Figure 3 on page 6.

Comments 3: Figure 5-6. Confidence intervals are also not shown on the graphs.

Response 3: We agree with this comment. We have changed the Figures 5 and 6 on pages 7-8.

Comments 4: Table 2. Here, authors must also provide confidence intervals for measured values, including for EDS analysis.

Response 4: The number of experiments we have done is relatively small (between 5 and 10). The results obtained are very close and within the measurement error. Despite the close results obtained, the confidence interval and, accordingly, the margin of error are large due to the relatively small number of experiments. Of course, the confidence interval can be reduced if we assume a smaller confidence level. For example, 75% instead of 95%. - Experimental part, page 5, paragraph 1.

Comments 5: Figure 7. The captions for the figures are misaligned.

Response 5: The remark has been taken into account and the figures caption of Figure 7 are corrected.

Comments 6: Figure 7. SEM images of 3D PETG with varying degrees of filling density of the inner layers and electroless deposed: (a-c) Ni-P coatings; (d) Cu coating.” Here, authors must also identify captions (a)-(c) for fill densities of 20%, 50%, and 100%.

Response 6: The remark has been taken into account and the figures caption of Figure 7 are corrected.

Comments 7: Figures 9, 10. There are no scale marks on SEM images of microstructures.

Response 7: We agree with this comment. We have changed the Figures 9 and 10 on page 11.

Reviewer 3 Report

Comments and Suggestions for Authors

The Authors present an interesting work on the metallic coating of 3D printed PET and PETG samples using various infill densities and etching parameters. The manuscript is well-written and easy to follow. The experimental conditions are well described, and the results are clearly presented. There are only some minor suggestions to improve the quality of the manuscript:

-In the introduction part, it is suggested to present more information from the literature about the metallic coating of 3D printed specimens. Numerous articles have been written on this topic.

-In Figure 1. b. the presented schematic of an FDM 3D printer with two extruders capable of composite material printing. The reference of this figure part is also about composite 3D printing. Did the Authors use dual-extruder FDM printer? If not, it is suggested to use the schematic of a single-extruder FDM printer.

-The type of FDM printer is not told by the Authors. Is it a custom-built model?

-Table 1. : The stoichiometric constants should be lower index numbers.

-It is suggested to include some macrophotographs of the coated samples.

-

Author Response

Comments 1: In the introduction part, it is suggested to present more information from the literature about the metallic coating of 3D printed specimens. Numerous articles have been written on this topic.

Response 1: We agree with this comment. [With the advent of rapid prototyping, metallization on polymers is attracting attention in a variety of technological applications, from electronic devices and automobiles to prosthetics. Green technologies exist that are able to metallize selected areas of 3D printed samples by introducing PdCl2 as a catalytic precursor into the polymer filament for surface activation and subsequent metallization. The methods were also developed for electroless metal deposition on polymer with Cu particles embedded in it [12-15]. – Introduction, page 2, paragraph 3

12. Akin, S.; Nath, Ch.; Jun, M.B. Selective Surface Metallization of 3D-Printed Polymers by Cold-Spray-Assisted Electroless Deposition, ACS Appl. Electron. Mater, 2023, 5 (9), pp. 5164–5175.; 13. Zhan, ; Tamura, T.; Li, X.; Ma. Z.; Sone, M.; Yoshino, M.; Umezu, S.; Sato, H. Metal-plastic hybrid 3D printing using catalyst-loaded filament and electroless plating, Additive Manufacturing, 2020, 36, 101556.; 14. Sharifi, ; Paserin, V.; Fayazfa, H(Ramona). Sustainable direct metallization of 3D-printed metal-infused polymer parts: a novel green approach to direct copper electroless plating, Advances in Manufacturing, 2024, 12, pp. 784–797.; 15. Siddikali, P.; Rama Sreekanth, P.S. Performance Evaluation of CNT Reinforcement on Electroless Plating on Solid Free-Form-Fabricated PETG Specimens for Prosthetic Limb Application, Polymers, 2022, 14(16), 3366.]

Comments 2: In Figure 1. b. the presented schematic of an FDM 3D printer with two extruders capable of composite material printing. The reference of this figure part is also about composite 3D printing. Did the Authors use dual-extruder FDM printer? If not, it is suggested to use the schematic of a single-extruder FDM printer.

Response 2: We agree with this comment. We have changed the Figure 1 on page 3.

Comments 3: The type of FDM printer is not told by the Authors. Is it a custom-built model?

Response 3: Agree. We have added the type of FDM printer - SKYMAKER A2 on page 3.

Comments 4: Table 1.: The stoichiometric constants should be lower index numbers.

Response 4: We agree with this comment. We have changed the stoichiometric constants in Table 1.

Comments 5: It is suggested to include some macrophotographs of the coated samples.

Response 5: The photographs of electroless metallized Ni-P and Cu coated on 3D samples are given in the graphical abstract. According to the requirements of the journal, the same photos cannot be given in the text.

Round 2

Reviewer 1 Report

Comments and Suggestions for Authors

This paper is good for publication

Reviewer 2 Report

Comments and Suggestions for Authors

The authors significantly corrected and improved the manuscript. I think the manuscript can be accepted for publication.